# Prevalence of Value-Based Care for Older People with Dementia Likely to Be Nearing End of Life: A Hospital Retrospective Cohort

**DOI:** 10.3390/healthcare12232382

**Published:** 2024-11-27

**Authors:** Ruzanna Shah, Danielle Ní Chróinín, Jenny He, Magnolia Cardona

**Affiliations:** 1Liverpool Hospital, Liverpool, NSW 2170, Australia; danielle.nichroinin@health.nsw.gov.au; 2Faculty of Medicine & Health, The University of New South Wales (UNSW Medicine & Health), Sydney, NSW 2033, Australia; jenny.he@student.unsw.edu.au; 3School of Population Health, The University of New South Wales, Sydney, NSW 2033, Australia; magnolia.cardona@unsw.edu.au; 4Institute for Evidence Based Healthcare, Bond University, Robina, QLD 4226, Australia

**Keywords:** dementia, end of life, advance care planning, polypharmacy, deprescribing, older adult

## Abstract

**Background**: Patients with dementia (PwD) nearing end of life (nEOL) do not always receive optimal end-of-life care, including timely specialist palliative care input. In hospitalized PwD likely to be nEOL, we aimed to determine the prevalence of goals of care discussions; the incidence and timing of referral to palliative care; factors associated with palliative care referral and timely (within 2 days) palliative care referral; and the prevalence of polypharmacy (>5 medications) and in-hospital deprescribing (cessation). **Methods**: A retrospective chart review of a cohort of PwD admitted under geriatric medicine 1 July 2021–30 June 2022 was conducted, screening to identify nEOL status. **Results**: A total of 298 patients (mean age of 83.5 [SD 7.4] and 51.3% females) were included in the final analysis. Eleven percent of eligible patients (33/298) died during admission. Overall, 80.9% had discussed an advance care plan (ACP). The mean time from admission to the discussion of an ACP was 1 day (SD = 5.02). One in twenty (5.4%) had their goals of care revised during admission, with 15 transitioning to palliation. Only 7.1% were referred to palliative care during admission. The mean time to referral was 9.8 days (SD 7.3; range 0–26). One in fourteen (7.4%) were discharged from hospital on an end-of-life pathway. In multivariable analysis, both the clinical frailty score (CFS) (aOR per unit increase 3.66; 95%CI 1.65–8.09, *p* = 0.001) and meeting ≥ 2 deterioration criteria (per CriSTAL tool) (OR 3.68; 95% CI 1.07–12.70, *p* = 0.039) were independently associated with referral to palliative care. Polypharmacy was common at admission (76.2%), with a mean number of medications of 8.4 (SD = 4). The median number of medications ceased during admission was two (IQR 0–4). **Conclusions**: Contrary to our hypothesis, we found a high quality of care of PwD likely nEOL, reflected by frequent ACPs in hospital, but it fell short of palliative care specialist input. Polypharmacy on admission and discharge may be further improved by aligning medication use with goals of care among PwD approaching the end of life, and considering the hospital nurse-driven identification of practice gaps is encouraged.

## 1. Introduction

Dementia is the seventh leading cause of death worldwide [1] and in Australia was the leading cause of death (accounting for 13% of all deaths) among women in 2021 [2]. Although it is well recognized that palliative care is recommend for life-limiting illnesses like dementia, patients with dementia (PwD) do not always receive optimal end-of-life care [3], due to multiple factors including, but not limited to, the under-recognition of dementia as a progressive, life-limiting illness and its palliative care needs by physicians, prognostic uncertainty, communication challenges, and resource limitations [4]. Guidance on ways to improve the delivery of high-quality end-of-life care for PwD is available in Australia and globally [5,6]. In Australia, the default management of acute exacerbations of terminal illness for older people is hospitalization rather than referral to palliative care. One reason is the minimal availability or limited subsidy of community and home-based palliative care in the health system, particularly for remote residents [7]. Hence, much remains to be done in the delivery of high-quality palliative care across health systems globally and at the national, district and point-of-care levels [8].

In our hospital, criteria for admission from the emergency department under geriatric medicine to the acute aged care ward are broad and include age-associated conditions such as delirium, dementia with behavioral problems, malnutrition associated with age-related conditions such as dementia and frailty, deconditioning and/or functional decline, gait abnormality, recent falls, and uncomplicated fragility fractures. Criteria for admission from the emergency department under palliative care includes if palliation was required for symptom control and the patient was registered under the South Western Sydney Local Health District (SWSLHD) Palliative Care Service. The Palliative Care Service also provides a referral-based consultative service for inpatients admitted under other services who require specialist palliative care input in relation to symptom control or end-of-life care.

While the need to address these barriers has been established, there is a paucity of clinically relevant evidence on characteristics of PwD that may predict the timely transition to palliative care or another conservative end-of-life care pathway in the current literature.

In an acute hospital setting, timely discussions about goals of care are crucial components of quality care in PwD, ensuring that their personal preferences and values are respected [9]. Advance care planning is a dynamic process that involves person-centered discussions about the person’s values and the type of health care they would want to receive if they became seriously ill and unable to express their preferences [10]. Early advance care planning may improve likelihood of PwD receiving care consistent with their preferences, thereby improving care outcomes for PwD and their families [11]. When patients with advanced dementia have lost decision-making capacity, healthcare proxies or legal guardians must intercede for them and make decisions that align with their known preferences and best interests [12]. PwD nearing the end of life (nEOL) are likely to benefit from clearly documented discussions regarding their advance care plan (ACP) [13] to guide treating clinicians in providing care that maximizes quality of life and death by minimizing aggressive interventions that have limited or no benefit.

However, timely discussions regarding ACPs for PwD nEOL can be challenging due to clinicians and carers not being aware of dementia as a life-limiting illness [14]. Other contributory factors to the suboptimal uptake of timely discussions regarding ACPs may include inherent uncertainty in illness trajectory and unpredictable prognosis, cultural and religious beliefs that can influence attitudes towards death and end-of-life care, and a lack of standardized protocols for conducting these important conversations [15].

Medication management is a key feature of aligning treatment with patient goals and preferences and another crucial component of quality care of PwD. Deprescribing in these patients involves the systematic reduction or cessation of medications that may be deemed unnecessary, ineffective, or potentially harmful. It aims to improve quality of life by reducing burdensome polypharmacy, adverse drug reactions (ADRs) and caregiver distress, thereby improving overall care outcomes [16,17].

In PwD likely to be nEOL, we aimed to determine the following:The extent of the discussion and documentation of goals of care and incidence and the timing of referral to palliative care.Factors associated with palliative care referral and timely (within 2 days of admission) palliative care referral.The prevalence of polypharmacy (as a marker of potential over-treatment) and the incidence of in-hospital deprescribing among these patients.

In this context, we hypothesized that (1) the current quality of care of PwD nEOL is likely to be suboptimal, with a low incidence of timely discussion and documentation of goals of care and referral to specialist palliative care, and (2) these PwD are likely to experience polypharmacy and suboptimal rates of in-hospital medication deprescribing.

## 2. Materials and Methods

### 2.1. Study Design

We conducted a retrospective observational cohort study including all patients with a diagnosis of dementia who were admitted under geriatric medicine for at least one night for one year between 1 July 2021 to 30 June 2022, and who met the eligibility criteria regardless of discharge outcome and location. For meaningful statistical analysis, we considered a minimum of 10 events per the Criteria for Screening and Triaging to Appropriate aLternative care (CriSTAL tool) item as ideal, and therefore, one year of data would be sufficient to provide a sample of around 300.

### 2.2. Inclusion Criteria

Patients aged ≥65 years.Patients with a confirmed diagnosis of dementia, as evidenced by the International Classification of Diseases (“ICD-10AM”) [18] code in their clinical record or discharge summary.Patients with a CriSTAL score of ≥6 as per the Criteria for Screening and Triaging to Appropriate aLternative Care (CriSTAL) tool to identify patients who were at high risk of death within the 3 months following admission [19].

The CriSTAL screening tool used in this study (freely available from the developers) has been prospectively validated for use across emergency departments in predicting short-term mortality in older patients after emergency department admission in Australia and Denmark [20]. A more recent randomized controlled trial in three Queensland hospitals used CriSTAL to identify risk of death in hospitalized older patients and examine clinicians’ actions in response to this risk knowledge [14,21]. They validated CriSTAL against the Supportive and Palliative Care Indicators Tool (SPICT).

### 2.3. Exclusion Criteria

Patients with more than two missing CriSTAL parameters.Patients with unclear or missing data on discharge disposition.

### 2.4. Data Extraction and Population

For patients who had multiple admissions, only the latest admission within the study period was considered.

Data were extracted on patient demographics, dementia diagnosis, frailty status using the Clinical Frailty Scale (see [App appB-healthcare-12-02382]), the presence of other key CriSTAL parameters (see [App appA-healthcare-12-02382]), evidence of specialist palliative care referral and involvement, time from admission to palliative care referral, and whether the patient was discharged from hospital on an end-of-life pathway. In relation to quality of care pertaining to advance care planning and medication management, we collected data on the number of medications on admission and discharge from hospital, the number and type of medications that were newly prescribed or ceased at the time of discharge from the hospital, the presence or absence of any discussions regarding goals of care, the person responsible for initiating the discussion regarding patient goals of care, evidence of advance care plans (ACPs) or pre-existing Advance Care Directives, whether the patient goals of care were revised during the admission, and the output format of the discussion.

The outcomes assessed included the following:The extent of the discussion and documentation of goals of care and the incidence and timing of palliative care referrals/residential aged care facility (RACF) referrals for end-of-life care.The factors associated with palliative care referral and timely (within 2 days of admission) palliative care referral, defined as presence of at least two of the following occurring within 2 days of admission: palliative care consultation/referral, initiation, or revision of patient goals of care during admission; issuing of limitations of treatment; deprescribing, and discharge to home/RACF on an end-of-life pathway.The prevalence of polypharmacy (defined by the World Health Organization as the concurrent use of >5 medications, [22]) and the incidence of in-hospital deprescribing.

People from culturally and linguistically diverse (CALD) backgrounds are defined as people born overseas in countries where English is not the main language spoken—that is, people whose country of birth is not Australia and its external territories, New Zealand, the United Kingdom, Ireland, the United States of America, Canada, or South Africa. This selection of countries is based on the main countries from which Australia receives overseas settlers who are likely to speak English, or people born in Australia whose main or preferred language spoken is not English [23]. Patients who died in hospital were excluded from the deprescribed-by-discharge counts.

### 2.5. Statistical Analysis

Statistical analyses were performed using Stata V13.0 (Stata Corp, College Station, TX, USA). We performed a descriptive analysis of patient characteristics to address primary outcome 1 and 3. Univariate and multivariate regression analyses were planned and used to evaluate variables potentially associated with palliative care referral and timely (within 2 days of admission) palliative care referral. Variables were included in the multivariate model if an association between a variable and outcome was observed in the univariate analysis (*p* < 0.05). Explanatory variables in the base logistic regression model included age, sex, and residential aged care status. Other variables examined were frailty (CFS) and ≥2 rapid response criteria present on admission.

### 2.6. Informed Consent

Patient consent was waived as the study was a retrospective observational study and de-identified information was used in the study. No patients or next of kin were contacted for the study. This was approved by the district Human Research Ethics Committee (see the end of this manuscript).

## 3. Results

### 3.1. Demographic Profile and CriSTAL Parameters

During the study period, 394 patients with a diagnosis of dementia were admitted to the study hospital mostly via the emergency department. Amongst 325 patients who met the age and diagnostic eligibility criteria for the study, 298 had a CriSTAL score ≥ 6 and were included in the final analysis. No eligible patients were excluded from the study.

Table 1 summarizes the demographic details and CriSTAL parameters of our patients. This indicates a very old and ethnically diverse, high-risk cohort with high levels of frailty and cognitive impairment and a recent history of hospitalizations and fall events.

### 3.2. ACP Discussions

Thirty-three patients (11.1%) died during admission. The vast majority had discussed an advance care plan (ACP) within a day on average, mostly with specialists about ward-based care, and one-third had a pre-existing ACP on presentation to hospital (Table 2). The majority (238/298; 80.1%) had plans formalized during the hospitalization using a hospital resuscitation form, which is not as comprehensive as the ACP form.

### 3.3. Incidence and Timing of Palliative Care Referrals

Only 21 (7.1%) patients were referred and reviewed by the inpatient specialist palliative care team during admission, and only 4/21 patients were referred in a timely (within 2 days of admission) manner. The mean time to referral was 9.8 days (SD 7.3; range 0–26). A small number of patients (22/298; 7.4%) were discharged from hospital on an end-of-life pathway, and of them, 10 were reviewed by the specialist palliative care team prior to discharge.

### 3.4. Factors Associated with Transition to Palliative Care

Numbers were insufficient to perform logistic regression for timely palliative care referral (within 2 days). Hence, our regression outcome was any palliative care referral. On univariate analysis, only CFS (OR 3.83; 95%CI 1.94–7.56, *p* < 0.001) and meeting ≥ 2 deterioration criteria (per CriSTAL tool) (OR 5.87; 95%CI 1.88–18.34, *p* = 0.002) were associated with a likelihood of palliative care referral. None of polypharmacy, ACP, age, sex, CALD status, and RACF residence were associated with palliative care referral (all *p* > 0.1). In multivariable analysis, in the final model, adjusting for age, sex and RACF residence, as per our a priori strategy, both CFS (aOR per unit increase 3.66; 95%CI 1.65–8.09, *p* = 0.001) and meeting ≥ 2 deterioration criteria (per CriSTAL tool) (OR 3.68; 95%CI 1.07–12.70, *p* = 0.039) were independently associated with referral to palliative care. Table 3 shows the final model. The direction of findings was unchanged when non-significant variables were removed from the model.

### 3.5. Prevalence of Polypharmacy and Incidence of In-Hospital Deprescribing

As shown in Table 4, amongst 298 patients, 227 (76.2%) experienced polypharmacy at admission, with 8.4 being the mean number of medications (SD = 4.0). Deprescribing was uncommon, occurring in 60/298 (20%) of patients. The median number of medications ceased during admission was two (IQR 0–4), with 31 different drug categories represented amongst 868 medications ceased. The most frequently ceased medications were antihypertensives (62/298, 20.8%), analgesics (41/298, 13.8%) and aperients/lipid-lowering drugs (both 36/298, 12.1%). The median number of new medications prescribed by discharge was two (IQR 0–4), with a total of 217/298 (72.6%) receiving new medications by discharge. Indications for these comprised 174 (58.4%) acute illness, 120 (40.3%) chronic illness and 23 (7.7%) palliative care/comfort. A total of 225 (75.8%) patients had polypharmacy on discharge (mean 8.3 medications; SD = 4.8). Polypharmacy on discharge was more common in those prescribed new medications (*p* < 0.001), including for any of the above indications (all *p* < 0.05), but was not associated with age or sex.

Out of the 62 patients who had their antihypertensives deprescribed, the indication for 50 patients was for hypotension from hypovolemia caused either by sepsis or dehydration and associated nephrotoxicity; however, for eight patients, it was for symptomatic orthostatic hypotension, and four patients were dying and no longer needed it. Moreover 37 patients had their analgesics deprescribed due to excessive sedation/hypoactive delirium. Out of the 36 patients (12.1%) who had their anti-lipids deprescribed, 27 patients had statin-related muscle injury, 5 patients had synthetic liver dysfunction, and 4 patients were dying in hospital. Out of the 41 patients (13.8%) who had aperients deprescribed, 35 patients experienced diarrhea and 6 patients were dying in hospital. In our study, only 61/298 (20.5%) patients had medication reconciliation led by a ward pharmacist.

## 4. Discussion

In this retrospective cohort study of 289 consecutive older people living with dementia and likely to be nearing end of life, we found that most patients (four in five) had discussions regarding their ACP during their hospitalization within a day of admission on average, highlighting a proactive approach by clinicians in initiating these crucial conversations. The prevalence of ACP documentation in our study was higher compared to earlier published studies with prevalence of ACP documentation between 0 and 5% [24,25]. The prevalence of these discussions was also higher in this 2021–2022 study than in an interventional study conducted over the early pandemic period of 2020–2021 in another Australian state. Those authors reported a reduction in ACP discussions and no change in palliative care referrals despite nudging the admitting team about patients’ elevated risk of death [14]. Internal hospital policies related to early discharge during the COVID-19 pandemic in the earlier study could explain this difference.

To our knowledge, our study is the first to report emergency department physicians and geriatricians emerging as the primary initiators of ACP discussions in aged care patients, highlighting the role of specialized medical personnel in addressing end-of-life care, although the focus was on ward-based care. This finding potentially gives way to consideration of new models of nurse-driven EOL discussions on wards in the Australian health system to fill the gap in the identification of polypharmacy and missed opportunities for palliative care referral. Additionally, the formalization of ACPs using hospital resuscitation forms was common, highlighting a proactive approach to documenting patient preferences regarding resuscitation and end-of-life care. Another new contribution to our knowledge was that among the multiple risk factors for death investigated, both CFS (per unit increase) and meeting ≥ 2 criteria for clinical deterioration (per CriSTAL tool) emerged as being strongly associated with palliative care referrals. This calls for a higher awareness of the overall risk of death among clinicians holding end-of-life discussions with patients who might benefit from a transfer to conservative management.

In our study, over one-third of patients deemed likely nEOL were residents of aged care facilities, underscoring the importance of timely discussions surrounding ACPs and end-of-life care preferences in aged care. Of the 33 patients who died, 15 had their goals of care revised to palliation from a more ‘active’ treatment plan because they had clinically deteriorated during their admission, reflecting a willingness amongst clinicians and patient/carers to adapt care plans based on changing clinical circumstances and patient preferences, leading to appropriate end-of-life care in hospital.

Despite the high uptake of EOL discussions identified and the known importance of referral to palliative care services in PwD likely nEOL [5], we found that only a small percentage of patients with dementia likely nEOL (7%) was referred during admission. This is consistent with another US study that found that 17% of PwD were referred to specialist palliative care during admission [26]. A slightly better outcome (but still suboptimal) was found in another cohort study that followed up older patients in Australian rural hospitals. The authors found that 31% of the deceased had used palliative care services after discharge in the last trimester of their life, despite having been deemed at risk of death in the emergency department [27].

The low rates of referral to palliative care in PwD may be due to lack of standardized criteria in non-cancer patients for referral to palliative care [28] or the expertise of the treating aged care teams in managing the end-of-life care in their patients, particularly those with less complex symptoms or behavioral disturbances related to their comorbidities and may be managed satisfactorily by an aged care team. In our study, PwD who were frail (CFS ≥ 5) were more likely to be referred to palliative care in a timely manner. This may be because those who are obviously frail are recognized by clinicians to experience adverse health outcomes, including poorer prognosis and higher mortality rates, and have a more complex symptom burden and higher care needs [29]. Our finding is consistent with other published studies that also identified an association between greater functional impairment in PwD with an increased likelihood of receiving palliative care [30,31]. Additionally, a systematic review found that an increased dependency of activities of daily living predicted poor prognosis and would benefit from involving specialist palliative care [32].

The mean time to specialist palliative care referral was longer than expected, compared to a study in USA where referrals to palliative care assessment were delayed for 2 days for older patients admitted through the emergency department [33]. This could be due to either difficulty in recognizing the point of unmet needs, the reluctance of the family/carer to accept palliative care, or variability in the threshold for referral to specialist palliative care by the primary treating team [34]. Timely specialist palliative care input plays a crucial role in improving the quality of care for PwD nEOL, as well as in supporting their families through the end-of-life journey. Specialist palliative care input may offer specific benefits over and beyond traditional end-of-life care compared to other specialties, such as managing persistent or refractory distressing symptoms and negotiating difficult discussions between the medical team and families surrounding the goals of treatment [35]. Additionally, specialist palliative care teams can coordinate ongoing end-of-life care for patients who choose to die at home by providing home-based clinical and related support services or direct hospice admission, thereby avoiding emergency department presentation.

In this study, the prevalence of polypharmacy in PwD likely nEOL at admission was high, with three-quarters of patients experiencing polypharmacy upon hospital admission and continuing through to discharge. Although not satisfactory, this finding was consistent with other Australian and international studies in developed nations that also reported PwD being at higher risk of polypharmacy than other older community dwellers [36,37,38]. The widespread problem of polypharmacy in PwD persists notwithstanding the awareness of its implications on quality of life, symptom burden and enhanced risk of hospitalization and death [39] and despite the published recommendations for its reduction [40]. This persistent finding can inform points for intervention.

Although one out of three patients had no medications deprescribed, many patients had one or more medications deprescribed during admission but also required the prescription of new medications, resulting in polypharmacy at discharge. Polypharmacy poses several potential problems for PwD nEOL and caregivers, including an increased risk of adverse drug reactions (ADRs), which may be severe and lead to further complications including injurious falls, confusion or drowsiness, reduced quality of life and increased burden on caregivers.

The impetus for deprescribing antihypertensives was to prevent adverse drug reactions (ADRs) rather than alleviate polypharmacy. Further, we found that that over 70% of patients received new medications by discharge. These new medications were prescribed across the gamut of acute illness, chronic illness, or palliative care/comfort indications. While the initiation of new medications may be warranted for the management of new acute illnesses or chronic conditions, it also contributes to the persistence of polypharmacy, potentially exacerbating ADRs and drug interactions [41]. Importantly, as dementia reaches its end stages, the ongoing treatment of chronic comorbidities may have limited prognostic benefit and may not significantly enhance the patient’s remaining life expectancy nor improve overall health outcomes. Thus, the emphasis needs to shift from managing chronic comorbidities to providing palliative care intervention that enhances comfort and preserves dignity in PwD nEOL.

### 4.1. Strengths and Limitations

Our study has several strengths, including the inclusion of a full sample of consecutively admitted patients, the use of a validated tool for identifying a subpopulation of hospitalized older patients with a short-term risk of death, and comprehensive data collection. This study has identified that frailty was associated with a timely transition to palliative care or another conservative end-of-life care pathway in PwD nEOL, and it also highlighted potential gaps that may be practically addressed by treating clinicians.

We acknowledge the limitations inherent in the retrospective, single-institution nature of this study, which may limit its generalizability. We relied on documentation that was available and acknowledge that it may not have always comprehensively reflected what was discussed. Not all patients may have had the number of medications on admission captured accurately in the absence of a thorough a medication reconciliation between a medical officer or pharmacist and their caregiver or GP, so there may be under- or over-estimations of medication numbers, and assessing the appropriateness of indication was beyond the scope of this study. We did not use a comorbidity index, and frailty is strongly associated with comorbidity [42]. Our study did not follow up with patients after discharge, so our mortality estimates were confined to the hospital stay. We are unable to report complete mortality figures within 3 months of admission for those who left hospital. We relied on a documented diagnosis of dementia, noting that this is often under-captured [43]. Lastly, we were unable to assess patient or carer outcomes/experiences, particularly in relation to experiences of care with palliative care input versus without, and the qualitative exploration of the lived experience would add depth to these findings [44].

### 4.2. Practical Implications

To fill the 20% gap in goals of care discussions in hospital clinical practice, clinicians may consider routinely utilizing available tools such as the CriSTAL score. To guide their review of potentially inappropriate medications/prescriptions in older adults, taking remedial action using the Screening Tool of Older Persons’ Prescriptions (STOPP) and Screening Tool to Alert to Right Treatment (START) criteria can minimize the polypharmacy prevalence at discharge. Decision trees are also available to guide clinicians in deprescribing high-risk medications (e.g., benzodiazepines) among older adults [45]. Targeted awareness education on the practice gaps may further enhance the end-of-life identification and management that aligns with patient wishes [46].

## 5. Conclusions

Overall, ACP discussions and documentation were more common than we anticipated in hospitalized people with dementia likely near the end of life, although the practice fell short of referrals to palliative care. Further, polypharmacy on admission and discharge from hospital were still prevalent in this subpopulation, and the quality of care may be further improved by aligning medication use with goals of care and implementing proactive inpatient medication review processes that consider the appropriateness, effectiveness, and safety of all medications prescribed to hospitalized patients. Given that the main initiators of EOL discussions are specialists in emergency departments, engaging nurses in filling the gap in ward care merits consideration. Further qualitative analysis of reasons for the low utilization of the palliative care pathway will inform points for interventions to close the gap in quality end-of-life care.

## Figures and Tables

**Table 1 healthcare-12-02382-t001:** Demographic details and CriSTAL parameters (n = 298).

Variables	Distribution
Male, N (%)	145 (48.7)
Female, N (%)	153 (51.3)
Age, years, mean (SD)	83.5 (7.4)
CALD background, N (%)	
Yes	195 (65.4)
No	103 (34.6)
Overall CriSTAL score, mean (SD)	8.2 (1.6)
Admission via Emergency Department, N (%)	
Yes	298 (100)
No	0 (0)
RACF, N (%)	
Yes	109 (36.6)
No	189 (63.4)
Frailty score (CFS) [[App appB-healthcare-12-02382]], mean (SD)	6.3 (0.8)
Frailty (CFS ≥ 5), N (%)	
Yes	297 (99.7)
No	1 (0.3)
≥2 RRT criteria on admission [[App appA-healthcare-12-02382]], N (%)	
Yes	19 (6.4)
No	279 (93.6)
Other risk factors/predictors:	
Advanced malignancy, N (%)	
Yes	35 (11.7)
No	263 (88.3)
Chronic kidney disease, N (%)	
Yes	35 (11.7)
No	263 (88.3)
Chronic heart failure, N (%)	
Yes	35 (11.7)
No	263 (88.3)
Chronic obstructive pulmonary disease, N (%)	
Yes	30 (10.1)
No	268 (89.9)
New cerebrovascular disease, N (%)	
Yes	20 (6.7)
No	278 (93.3)
History of or new myocardial infarction, N (%)	
Yes	15 (5.0)
No	283 (95.0)
Moderate/severe liver disease, N (%)	
Yes	7 (2.3)
No	291 (97.7)
Cognitive impairment, N (%)	
Yes	298 (100)
No	0 (0)
Proteinuria on a spot urine sample, N (%)	
Yes	4 (1.3)
No	0 (0)
Do not know	294 (98.7)
Abnormal ECG, N (%)	
Yes	67 (22.5)
No abnormality	231 (77.5)
Hospitalization within last 1 year, N (%)	
Yes	197 (66.1)
No	101 (33.9)
ICU admission within last 1 year, N (%)	
Yes	3 (1.0)
No	295 (99.0)
Fall within last 3 months, N (%)	
Yes	144 (48.3)
No	154 (51.7)

CALD = culturally and linguistically diverse (people from CALD backgrounds are defined as people born overseas in countries where English is not the main language spoken—that is, people whose country of birth is not Australia and its external territories, New Zealand, the United Kingdom, Ireland, the United States of America, Canada, or South Africa. This selection of countries is based on the main countries from which Australia receives overseas settlers who are likely to speak English, or people born in Australia whose main or preferred language spoken is not English); CriSTAL = criteria for screening and triaging to appropriate alternative care; RACF = residential aged care facility; CFS = clinical frailty scale; RRT = rapid response team; ECG = electrocardiogram; ICU = intensive care unit; SD = standard deviation.

**Table 2 healthcare-12-02382-t002:** Prevalence and content of ACP discussions.

Variables	Distribution, N (%) Unless Otherwise Stated
ACP discussion	
Yes	241 (80.9)
No	57 (19.1)
Time from admission to ACP discussion, days, mean (SD), [range]	1 (5.0) [0–26]
Prior ACP	
Yes	91 (30.5)
No	207 (69.5)
Initiators of ACP discussion on admission	
Emergency department physicians	107 (35.9)
Geriatricians	106 (35.6)
Advanced trainees	27 (9.1)
GP	1 (0.3)
Goals of care discussed	
Ward-based care	201 (67.5)
ICU level care	13 (4.4)
Full resuscitation	9 (3.0)
Palliation	17 (5.7)
CPR on ward if shockable rhythm	1 (0.3)
Discharge to RACF for end-of-life care	
Yes	1 (0.3)
No	297 (99.7)
Goals of care revised during admission	
Transitioned to palliation	16 (5.4)
Died during admission	33 (11.1)
Output format of ACP discussion	
Hospital resuscitation form	238 (80.1)
Care of dying patient pathway form	1 (0.3)
Not documented	59 (19.6)

ACP = advance care plan; GP = general practitioner; ICU = intensive care unit; RACF = residential aged care facility; CPR = cardiopulmonary resuscitation; SD = standard deviation.

**Table 3 healthcare-12-02382-t003:** Factors associated with [any] palliative care referral.

Variable	Odds Ratio	95%CI	*p* Value
Age (per year increase)	1.01	0.94–1.08	0.823
Sex	2.47	0.85–7.21	0.097
RACF	0.96	0.35–2.69	0.945
CFS score (per point increase)	3.66	1.65–8.09	0.001
≥2 RRT criteria on admission	3.68	1.07–12.70	0.039

RACF = residential aged care facility; CFS = clinical frailty scale; RRT = rapid response team; CI = confidence interval.

**Table 4 healthcare-12-02382-t004:** Prevalence of polypharmacy and incidence of in-hospital deprescribing.

Variables	Distribution, N (%) Unless Otherwise Stated
Polypharmacy on admission	
Yes	227 (76.2)
No	71 (23.8)
Pharmacist-led medication reconciliation	
Yes	61 (20.5)
No	237 (79.5)
Number of medications on admission, mean (SD)	8.4 (4.0)
Deprescribing of ≥1 medications	
Yes	204 (68.5)
No	94 (31.5)
Number of medications ceased, median (IQR)	2 (0–4)
Polypharmacy on discharge	
Yes	225 (75.8)
No	73 (24.2)
Number of medications on discharge, mean (SD)	8.3 (4.8)
Number of new medications prescribed, median (IQR)	2 (0–4)
New medications prescribed by discharge	
Yes	217 (72.6)
No	81 (27.4)
New acute illness medications prescribed by discharge	
Yes	174 (58.4)
No	124 (41.6)
New chronic illness medications prescribed by discharge	
Yes	120 (40.3)
No	178 (59.7)
New palliative care/comfort medications prescribed by discharge	
Yes	23 (7.7)
No	275 (92.3)

IQR = interquartile range; acute illness medications = for treatment of an acute illness during admission; chronic illness medication = for treatment of known chronic illness; palliative care/comfort medications = for end-of-life symptom control; SD = standard deviation.

## Data Availability

The data that support the findings of this study are available from the first author upon reasonable request.

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
