# Peer review of "Prevalence of Value-Based Care for Older People with Dementia Likely to Be Nearing End of Life: A Hospital Retrospective Cohort"

_healthcare, 2024, doi:10.3390/healthcare12232382_

Round 1
Reviewer 1 Report
Comments and Suggestions for Authors
Thank you for providing the reviewer an opportunity to evaluate this manuscript, it was interesting. The reviewer has provided the following suggestions and comments.
1. In abstract, the reviewer suggested that including the gender and mean age of the sample would enhance the generalizability of the findings.
2. The authors should supplement the “quality health services and palliative care” issued by the World Health Organization (2021) at the national, district, and point-of-care levels in the introduction. The authors should incorporate the attributes of hospital-related services, including detailed of service teams and home services, in the service points. The readers will gain a better understanding of the services.
3. In the material and methods part, what motivated the authors to select this specific time period for data collection? What do authors consider to be the suitable sample size, determined by relevant statistical methodologies?
4. In the sample or group examined in the study including mortality, did the patient experience a good death? What is the author's definition of a good death?
5. The reviewer concurs with the authors' arguments and findings but should address the concerns mentioned by the reviewer.
Reviewer 2 Report
Comments and Suggestions for Authors
Thank you for your hard work.
I would like to inform you of the corrections.
1. Research method
Is this study a cohort study? The research method presented in the abstract and research method is indicated as cohort, but the research results are not the results of a cohort. Furthermore, the statistical analysis method is indicated as regression analysis, but the table and results are not presented that way. Is it a regression analysis? If it is a regression analysis, which table is the regression analysis? What is the result of that regression analysis? Tables 1 to 3 are all about n and %, and there are no results other than these. The abstract shows OR and CI values, but where is that information? Where is the table for 3.4. Factors associated with timely transition to palliative care? Isn't n and % by research period a survey rather than a study? So what does this study mean?
2. Table presentation
The table is difficult to read. Shouldn't the classification be presented before the variables in the table? The classification is not accurate. For example, in Table 1, n=195 people in CALD, where do the remaining 103 people belong? Please summarize Table 1, Table 2, and Table 3 again.
3. Discussion
The most important thing in this study is to derive the meaning of the accumulated results because it is a cohort study, and to present the basis and the researcher's opinion for this. However, this is not appropriate in this study. The basis and claims for all results should be derived by presenting previous studies, but the discussion was not properly described.
4. Conclusion
The conclusion of this study was presented, but the necessity and specialness of this study were not presented, and it is similar to existing studies and is ambiguous. Please present a conclusion based on the results of this study.
5. References
Out of the 41 references, 21 are not from the last 5 years. Please revise them to recent references. In addition, please check the notation method of the references and revise them to the description method of the journal according to the regulations of this journal.
Reviewer 3 Report
Comments and Suggestions for Authors
Dear authors:
I have reviewed your paper entitled “Prevalence of value-based care for older people with dementia likely to be nearing end of life: a hospital retrospective cohort”. It is a retrospective observational study that included patients with a diagnosis of dementia who were admitted under Geriatric Medicine, between 1st July 2021 to 30th June 2022.
This study aims to: to determine the prevalence of goals of care discussions, incidence and 15 timing of referral to palliative care, factors associated with timely palliative care/end-of-life pathway 16 and the prevalence of polypharmacy (>5 medications) and in-hospital deprescribing (cessation).
First, I would like to congratulate you on your work and submission. The paper is well-written, clear, methodologically sound, and particularly relevant to the field of palliative care. Additionally, you effectively address the issue of polypharmacy, a significant concern in today's healthcare landscape. As an external reviewer, I would like to offer the following suggestions and reflections for improvement:
· - Ethical issues: please insert the idea that was obtained authorization to use the Instruments used (Cristal); and please insert a paragraph explain that being a retrospective cohort study, it was not obtained informed consent.
· - In discussion: you could insert a point 4.1- Strengths and Limitations and 4.2- Practical implications
I have no further suggestions currently, and I wish you the best of luck with the publication of your paper! And I hope to ear more about your work.
Congrats, once again.
Best regards.
